# Enhancing Effect of Chloride Ions on the Autocatalytic Process of Ag(I) Reduction by Co(II) Complexes

**DOI:** 10.3390/ma13204556

**Published:** 2020-10-14

**Authors:** Loreta Tamašauskaitė-Tamašiūnaitė, Aldona Jagminienė, Ina Stankevičienė, Karolis Ratautas, Gediminas Račiukaitis, Eugenijus Norkus

**Affiliations:** Department of Catalysis, Center for Physical Sciences and Technology, Saulėtekio Ave. 3, LT-10257 Vilnius, Lithuania; aldona.jagminiene@ftmc.lt (A.J.); ina.stankeviciene@ftmc.lt (I.S.); karolis.ratautas@ftmc.lt (K.R.); gediminas.raciukaitis@ftmc.lt (G.R.); eugenijus.norkus@ftmc.lt (E.N.)

**Keywords:** electroless silver plating, Co(II) complexes, chloride ions

## Abstract

In this work, the possibilities of increasing the rate of electroless silver plating without a rise in the concentration of reactants or elevation of temperature were studied. The effect of halide additive, namely chloride ions, on the rate of electroless silver deposition was investigated, using conventional chemical kinetics and electrochemical techniques. It was found that the deposition rate of electroless silver increased 2–3 times in the presence of 10–20 mM of chlorides, preserving sufficient stability of the solution.

## 1. Introduction

Chemical deposition of silver layers (silver mirror formation) is the oldest known electroless plating process, but nowadays, the deposition of electroless silver is less effective and convenient compared to the electroless nickel or copper plating. Many reducers can be used for obtaining metallic silver from Ag(I) compounds, but generally, it is not so easy to get compact silver coatings due to the plenteous reduction of Ag(I) in the solution bulk. Unstable Ag(I)-ammonia (NH_3_) complex solutions with glucose, tartrate, formaldehyde, etc. have been used as reducing agents in plating for many years. The thickness of the coatings obtained from such solutions is not large and typically less than 1 μm [1,2].

More effective electroless silver plating solutions have been developed using Ag(I)-cyanide complex and amine boranes [3] or hydrazine [4] as reducing agents. At 40–50 °C, the deposition rate can reach 4 μm h^−1^, and in the presence of stabilizers, these solutions are sufficiently stable [1,4,5].

It has been reported that Co(II) complexes with NH_3_ can be applied as Ag(I) reducers [6,7,8]. Under some conditions, there is a significant predomination of the deposition of silver on the surface to be plated compared with that formed in the solution bulk. Silver plating systems of high stability are obtained, even in the absence of stabilizing additives, plating rate being as high as 1–3 μm h^−1^. The reducing agent Co(II) can be regenerated: its oxidation product, Co(III) species, is reducible by both chemical and electrochemical means. Thermodynamic and kinetic analysis has shown [8,9,10,11,12,13,14] that the rate of the overall process mostly depends on the amount of free NH_3_ concentration, which is mainly determined by the total concentration of ammonium species (NH_3_ + ${\mathrm{NH}}_{4}^{+}$) and the solution pH. The influence of all main factors: solution pH, temperature, concentrations of Ag(I), Co(II), and ammonium species, on the Ag deposition rate is determined.

It is worth knowing that according to Ref. [8], the electroless silver plating solutions are stable at a temperature of 50 °C, and the plating rate reaches ca. 3.2 μm h^−1^. Theoretical aspects of electroless silver plating using Co(II)-NH_3_ complexes are discussed in Refs. [9,11,12], showing that Co(NH_3_)_5_^2+^ and Co(NH_3_)_6_^2+^ complex species are involved in the electroless silver plating process. The reduction process is shown to proceed according to the stoichiometric Equation (1):(1)$$\mathrm{Ag}{\left({\mathrm{NH}}_{3}\right)}_{2}^{+}+\;\mathrm{Co}{\left({\mathrm{NH}}_{3}\right)}_{n}^{2+}\;\overset{\mathrm{Ag}}{\to }\;\mathrm{Ag}+\;\mathrm{Co}{\left({\mathrm{NH}}_{3}\right)}_{6}^{3+}+\;\left(n-4\right){\mathrm{NH}}_{3}$$
where n = 5 or 6.

The detailed kinetic study of electroless silver plating using Co(II)-NH_3_ complexes as a reducing agent is given in Refs. [10,11,13].

The silver layers obtained in the Ag(I)-Co(II)-NH_3_ system have a regular crystalline structure [15] with good anti-corrosion ability [16] and optical reflection characteristics [17].

Co(II) complexes with organic amines, e.g., 1,3-propylenediamine, can also be used as reducing agents in electroless silver plating solutions [18]; the plating process parameters are similar to those of the ammonia solutions.

At the high quality of silver coatings deposited by using an Ag(I)-Co(II) system and the high stability of the plating solutions, the silver plating rate may be too low for some practical applications where a thicker layer of silver is required. Alongside the practical importance, an acceleration of electroless silver deposition from the Co(II)-type solutions is a more general problem of enhancing the selective surface reaction of metal ion reduction without a considerable change in the process rate in solution bulk. It is agreed that the overall process of electroless metal consists usually of at least two interdependent electrochemical reactions: the anodic oxidation of the reducer and the cathodic reduction of the metal ions, taking place on the same catalytic surface. In our case, the anodic partial reaction—the oxidation of Co(II)—is sensitive to the ligands forming Co(II) complexes. It has been documented that the replacement of NH_3_ in a Co(II) complex by ethylenediamine (En) increases the anodic oxidation rate by a factor of 10–40 [18]. But the application of Co(II)-En complex for the electroless silver deposition is limited because of its too high reducing activity, leading to the fast reduction of Ag(I) in the volume of solution.

The anodic oxidation of Co(II)-ethylenediamine complex compounds on copper has been shown to be accelerated by halide ions, even at low concentrations [19,20]. A similar effect could be expected for other processes involving Co(II)-amine complexes. Therefore, this work aimed at the investigation of chloride ion effect on the process of electroless silver plating using Co(II)-NH_3_ complex compounds as a reducing agent in detail by electrochemical quartz crystal microgravimetry (EQCM), the method that was not used earlier when investigating electroless silver plating systems. The electrochemical partial reactions of this process were also investigated.

## 2. Experimental

### 2.1. Solutions

Electroless silver deposition solutions were prepared as follows: the mixture of (NH_4_)_2_SO_4_ and 25% NH_3_ solutions was added to the AgNO_3_ solution. The solution was deaerated with Ar for 15 min. After that, the CoSO_4_ solution was added. The main electroless silver deposition solution contained (M): AgNO_3_—0.04, CoSO_4_—0.10, (${\mathrm{NH}}_{4}^{+}$ + NH_3_)—4 or 7. The solution pH was regulated by changing the molar ratio of ${\mathrm{NH}}_{4}^{+}$ and NH_3_ (Table 1). Analytical grade chemicals and ultra-pure water with a resistivity of 18.2 MΩ cm^−1^ were used for preparing the solutions.

### 2.2. Electroless Silver Deposition

Silver coatings were deposited onto glass substrates with a geometric area of 7 cm^2^. At first, the glass substrates, degreased in an acidic Cr(VI) solution, were sensitized in a 2 g L^−1^ SnCl_2_ solution for 5 min, then rinsed with ultra-pure water and further activated in a 10 g L^−1^ AgNO_3_ solution, followed by rinsing with ultra-pure water. After that, the activated substrates were immersed in the deaerated electroless silver plating solutions (Table 1) at a temperature of 20 or 50 ± 1 °C for 30 min. Moreover, the reaction vessel was closed, and all vessel volume was filled with the solution to prevent the oxidation of Co(II) ions by atmospheric oxygen. The bath loading in kinetic experiments was 7 dm^2^ L^−1^.

### 2.3. Silver Coatings Characterization

The mass of the silver coating deposited on the glass substrate was determined by weighing, whereas the thickness of the one was calculated using the density of pure bulk silver. The atomic force microscope (AFM) TopoMetrix Explorer SPM (Veeco, Santa Clara, CA, USA) was used for the silver surface inspection.

### 2.4. Determination of the Real Surface Area of the Ag Electrode

The real silver coating surface (*S*_R_, cm^2^) was determined using the underpotential deposition (UPD) of the lead (Pb) monolayer on the Ag electrode surface, as described in [21]. Briefly, the measurements were carried out in a 0.1 M NaOH and 0.5 mM Pb(NO_3_)_2_ solution. At first, the silver oxide (Ag_2_O) was removed from the surface of the working Ag electrode (the same quartz crystal electrodes coated with silver, as described in Section 2.3, were used) by holding the potential at −0.10 V for 5 s. During a cyclic scanning of the potential in the range from 0 to −0.57 V at a scan rate of 50 mV s^−1^, a Pb monolayer was formed and dissolved. The typical cyclic voltammogram is presented in Figure 1. The charge (*Q*, μC) used for the anodic dissolution of the Pb monolayer was calculated by integration of the potentiodynamic curve obtained in the range from −0.45 to −0.2 V (Figure 1).

The *S*_R_ of the Ag electrode was calculated using the charge of Pb monolayer (*Q*_Pb_) necessary to form a monolayer on 1 cm^2^ of the electrode being equal to 280 μC cm^−2^ according to the Equation (2):*S*_R_ = *Q*/*Q*_Pb_(2)

The roughness factor *R*_n_ was calculated according to Equation (3):*R*_n_ = *S*_R_/*S*_G_(3)
where *S_G_* is the geometric area of silver coating.

The surface roughness was also calculated from the EQCM data. The deposited and dissolved Pb mass was obtained from the frequency change using the theoretical calibration constant 7.8 ng cm^−2^ Hz^−1^, and the surface area was calculated by using Pb monolayer mass 320 ng cm^−2^. The values of *S*_R_ of silver coating calculated from the coulometric and EQCM data were in satisfactory agreement. The more detailed description of the determination of the Ag real surface area by these techniques is given elsewhere [21].

All measurements were repeated at least three times, and a mean value was calculated.

### 2.5. Electrochemical and EQCM Measurements

The home-built electrochemical quartz crystal microbalance (EQCM) system employed was the same as that in [20,22]. Briefly, a three-electrode electrochemical cell was used. A working electrode was a gold-coated AT-cut quartz crystal of 6 MHz fundamental frequency (Intellemetrics Ltd., Paisley, UK) with a geometric area of 0.636 cm^2^, whereas an Ag/AgCl/KCl_sat_ electrode and a Pt-wire were used as a reference and a counter electrode, respectively. Cyclic voltammograms (CVs) were recorded from the open-circuit potential to the cathodic direction at a scan rate of 2 mV s^−1^. The values of the measured electrode potential, current, and frequency (with the stability of ±0.5 Hz) were transferred to the PC every 1.3 s. The quartz crystal frequency change rate (*df/dt*) was found as a difference between two frequencies measurements per 1 s and was used in this work for the calculation of the anodic/cathodic Ag current. The calibration constant 140 ± 10 Hz s^−1^ mA^−1^ was found in separate experiments in Ag(I)-NH_3_ solutions. Moreover, this value was close to that calculated from Sauerbrey’s [23] equation for the reduction of Ag(I) ions and corresponded to the sensitivity of EQCM used 7.98 Hz ng^−1^ (12.55 Hz ng cm^−2^), whereas a theoretical value of the sensitivity is 7.8 Hz ng^−1^ (12.26 Hz ng cm^−2^). Ag(I) reduction and Ag anodic dissolution currents were calculated by converting the *df/dt* to the current equivalent using this calibration constant. Partial currents of the oxidation of Co(II) and reduction of Co(III) were determined as a difference between the measured net current and that of Ag reduction/dissolution derived from EQCM data. The electrode potential was quoted versus a standard hydrogen electrode (SHE).

Before the measurements, a silver coating was electrodeposited onto a gold-coated quartz crystal substrate mounted in the cell from the 0.5 M AgNO_3_ and 0.5 M HNO_3_ solution at a current density of 4.7 mA cm^−2^ for 2 min.

The electroless silver coating was deposited onto the previously electroplated silver coating from an Ag(I)-Co(II)-NH_3_ solution for 5 min. The solution composition (M) was: Ag_2_SO_4_—5 × 10^−3^; CoSO_4_—0.1; NH_3_—4.0; pH 10.9. KCl additive concentration varied from 1 to 100 mM.

## 3. Results and Discussion

### 3.1. Electrochemical and EQCM Measurements

The kinetics of electroless silver deposition using the Co(II) complexes as reducing agents was investigated in detail. It was found that the addition of chlorides enhances Ag(I) reduction by Co(II) in NH_3_-containing solutions, as depicted in Figure 2. The accelerating effect of chloride ions is observed in a wide range of NH_3_ concentrations—in the entire practical pH range of electroless silver plating solutions of this type, beginning from pH 8.75 (0.5 M NH_3_) and concluding by the solution containing 3.8 M NH_3_, pH 11.2. Moreover, the chloride effect is seen at the concentration as low as 1 mM, and the silver deposition rate increases most strongly with a rise in chloride concentration until ca. 10 mM. At upper chloride concentrations, the silver plating rate increases only a little, and the dependences of “plating rate to Cl^−^ concentration” have a similar form in most cases, with some exception for the solution at pH 10.2 (Figure 2).

The acceleration of the autocatalytic silver deposition by chloride ions is relatively large: in the pH range 9.2–11.2, the addition of 10 mM KCl increases the plating rate from ca. 0.6 (chloride-free solution) to 1.2–1.5 μm 30 min^−1^ (Figure 3), and at higher chloride concentrations, the plating rate reaches 1.8 μm 30 min^−1^. So, the addition of 10 mM chloride leads to an acceleration of electroless silver deposition by a factor 2–2.5, and at higher chloride concentrations (40–50 mM), a three-fold acceleration is achieved.

The accelerating effect of chloride is also observed in autocatalytic silver deposition solutions with higher ammonia-ammonium (NH_3_ + ${\mathrm{NH}}_{4}^{+}$) salt concentration 7 M at a solution pH 11.05 (Figure 4).

When comparing the results of electroless silver deposition obtained at a temperature of 50 °C with those for 20 °C, the rate of silver deposition increases with an increase in temperature (Figure 5). At 50 °C, as seen in Figure 5, the rate also increases most strongly with a rise in chloride concentration until ca. 10 mM. At upper chloride concentrations, the silver deposition rate changes only a little but is larger than that at 20 °C.

### 3.2. Characterization

The roughness factor (*R*_n_) of the electroplated silver electrode, which is used as a substrate for electroless plating, is found to be 1.5. When using an electroless plating solution containing 10 mM additive of chloride, the electrolessly deposited silver coating has more developed surface roughness compared with the substrate—the surface roughness factor being 2.

The atomic force microscope (AFM) was used for the silver surface inspection. The typical surface profiles and images are shown in Figure 6. As evident, the surface morphology of silver coatings obtained with a solution containing a chloride additive differs from that obtained without a chloride additive (cf. Figure 6a,b). The dimensions of silver crystallites decrease, and the surface area increases visibly.

It is worth to notice that the adhesion of silver coatings is sufficient—the coatings obtained to pass the scotch tape test.

### 3.3. EQCM Investigations

To obtain further information, the effect of chloride ions for electroless silver deposition was investigated in more detail by EQCM. As mentioned before, the autocatalytic Ag(I) reduction by Co(II) in aqueous NH_3_ solutions (Equation (1)) is the sum of two (cathodic and anodic) partial reactions coinciding on the catalytic surface:(4)$$\mathrm{Ag}{\left({\mathrm{NH}}_{3}\right)}_{2}^{+}+e\;\to \mathrm{Ag}+2{\mathrm{NH}}_{3}$$

And
(5)$$\mathrm{Co}{\left({\mathrm{NH}}_{3}\right)}_{n}^{2+}+\;\left(6-n\right){\mathrm{NH}}_{3}\;\to \;\mathrm{Co}{\left({\mathrm{NH}}_{3}\right)}_{6}^{3+}+e,\;\left(n=5,\;6\right)$$

Figure 7 shows the dependence of open-circuit potential (a), frequency change rate (b), and change in quartz oscillator frequency (c) for electroless silver deposition from the plating solution containing the both Ag(I) ions and reducing agent Co(II) without chloride additive (solid line) and the one containing 1 (dashed line) and 10 mM (dash-dotted line) chloride additives. When the rates of both cathodic (Equation (4)) and anodic (Equation (5)) partial reactions are equal, the silver electrode attains the mixed potential (*E*_m_) value (Figure 7a). The *E*_m_ values are quite stable during the electroless silver deposition and are shifted to more positive potential values with the increase of the chloride concentration (Figure 7a). The autocatalytic silver deposition rate in all cases detected by EQCM is constant with time (Figure 7b), indicating a steady-state of the system. In all cases, the quartz crystal frequency decreases, e.g., the coating mass (thickness) increases constantly with time (Figure 7c). Furthermore, the rate of Ag(I) reduction by Co(II)-NH_3_ complexes depends on the concentration of chloride ions and rises with an increase in the chloride concentration up to 10 mM (Figure 7c). The chloride additive of 10 mM enhances the electroless silver deposition rate (Δ*f*) about ca. 3.6 times as compared with the additive-free solution (Figure 7c). The inset in Figure 7c shows the mass gain of silver. As evident, ca. 46 µg of silver could be obtained after 5 min deposition in the complete Ag(I)-Co(II)-NH_3_ electroless plating solution containing the 10 mM of chloride additive.

The QCM (quartz crystal microbalance) measurements, in parallel to the CV by determining the mass change, allowed to detect the rate of partial reactions (Ag(I) reduction, Ag dissolution, and Co(II) oxidation). Figure 8 shows the dependence of the reduction of Ag(I)-NH_3_ complexes on chloride ions concentration in the solution containing only Ag(I) ions. Simultaneous CV and QCM scans were started to record from the open-circuit potential to the cathodic direction. The stabilized *I*–*E* (solid line) and d*f*/d*t*–*E* (dashed line) dependencies show their shapes to be identical when the calibration constant of 140 Hz s^−1^ mA^−1^ was used (Figure 8). It is seen that in the presence of chloride ions, the reduction of Ag(I) is started at potentials ~0.16–0.17 V, and the limiting currents are reached at ~0.12 V (Figure 8b,c), whereas the limiting currents of the reduction of Ag(I) in the solution without chloride additive are observed at a more negative potential value of ~0.06 V.

Figure 9 presents the CVs recorded in the solution containing only Co(II) ions without (a) and with chloride additive of 1 (b), 10 (c), and 100 (d) mM. The insets (b’) and (d’) show the CVs at an anodic potential limit of 0.1 V. The dashed lines represent the anodic/cathodic Ag current. As evident, in the electrode potential range from −0.25 to 0 V, the calculated silver currents from QCM data is about 0 mA. This indicates that in this potential region, the measured current should be attributed to the Co(II)/Co(III) oxidation/reduction reaction (Figure 9). As evident from the data in Figure 9, the anodic Ag dissolution is detected at potentials more positive than 0 V, e.g., the Ag dissolution occurs simultaneously with the oxidation of Co(II). Furthermore, the rate of the oxidation of Co(II) is lower at the potentials more positive than 0 V of anodic Ag dissolution (Figure 9, the inset b’).

It should be noted that the process of Co(II) anodic oxidation on the Ag electrode is enhanced when chloride ions are presenting in solution. At 1 mM level up to 10 mM, KCl additive increases the rate of Co(II) oxidation only narrowly, while at 100 mM, KCl increases the rate considerably (Figure 9). The 100 mM KCl additive increases the Co(II) oxidation rate about four-fold. Acceleration of Co(II)-NH_3_ complexes oxidation in the presence of chloride is related to the adsorption of chloride ions on the silver surface, which are, possibly, involved in the oxidation of Co(II) and mediate the electron transfer. The enhancing effect of halide ions is observed during the oxidation of Co(II)-En complex on the copper electrode [20,22]. It has been shown that halide ions facilitate the electrons transfer through the halide bridge. The importance of halide-containing complexes in a bridged electron transfer occurring in the homogeneous redox reactions of the Co(III)/Co(II) couple is known [24].

CV and QCM measurements in the complete electroless plating solution, containing both reducing agent (Co(II)-NH_3_ complex) and metal ions to be reduced (Ag(I)-NH_3_ complex) in the presence of chloride ions, show more complicated electrochemical behavior (Figure 10) compared with the data obtained in Ag(I)-NH_3_ (Figure 8) or Co(II)-NH_3_ (Figure 9) solutions.

This would be expected for the situation when anodic and cathodic processes are coinciding on the same metal surface at the same potentials. The current measured corresponds to the algebraic sum of anodic Co(II) oxidation (and Ag dissolution at potentials positive enough) and cathodic Ag(I) reduction currents. The cathodic silver deposition (or silver dissolution) rate is obtained from EQCM data. The current of the anodic partial process of Co(II) oxidation is found from the difference between the current measured by CV and a partial current of cathodic reduction of Ag(I), or anodic dissolution of Ag calculated from the quartz crystal frequency change rate.

Thus, such partial currents of Ag(I) reduction and Co(II) oxidation can be compared with those measured in separate Ag/Ag(I)-NH_3_ (Figure 8) and Ag/Co(II)-NH_3_ (Figure 9) systems. Figure 11 shows the comparison of the Ag partial reactions measured in an Ag/Ag(I)-NH_3_ system without Co(II) from Figure 8 and that obtained in the complete Ag/Ag(I)-Co(II)-NH_3_ system from Figure 10. As illustrated in Figure 10 and evident from Figure 11, the rate of the cathodic partial process in the presence of Co(II) is similar, in general, to that observed in a separate Ag/Ag(I)-NH_3_ system. At chloride concentrations of 1 to 10 mM, limiting currents are observed at more positive potential values (<0.12 V) (cf. Figure 10b,c and Figure 11b,c) as compared with that in the additive-free solution (Figure 10a and Figure 11a). In the case of the electroless silver deposition solution without chloride ions, limiting current is not reached yet at these potentials.

The comparison of Co(II) oxidation and Co(III) reduction currents in the presence of Ag(I) (Figure 10) with those in a separate Ag/Co(II)-NH_3_ system (Figure 9) shows that in the presence of Ag(I), the Co(II) oxidation current has similar value as in the absence of Ag(I).

Chloride ions play an important role in the enhancement of the rate of electroless silver deposition, and its effect is observed in both anodic Co(II) oxidation and cathodic Ag(I) reduction partial reactions and could be explained by the formation of halide bridges.

## 4. Conclusions

In a study of an electroless silver deposition system with a reducing agent Co(II)-ammonia complexes, it was found that the addition of chlorides accelerates the overall process of electroless silver deposition. The addition of 10 mM chloride leads to an acceleration of electroless silver deposition by a factor of 2–2.5, and at higher chloride concentrations (40–50 mM), a three-fold acceleration is achieved.

The data of EQCM measurements confirm that the accelerating effect of chloride ions is attributed to the enhancement of both partial electrochemical reactions—mainly the anodic oxidation of Co(II) and, to a lesser extent, the cathodic reduction of Ag(I). It has been determined that the presence of chloride ions increases the anodic oxidation of Co(II)-NH_3_ complex, e.g., the 100 mM KCl additive increases the rate of Co(II)-NH_3_ complex oxidation about four-fold. The effect of chloride ions may be related to the adsorption of chloride ions on the silver surface and the formation of chloride bridges, which mediate the electron transfer.

## Figures and Tables

**Figure 1 materials-13-04556-f001:**
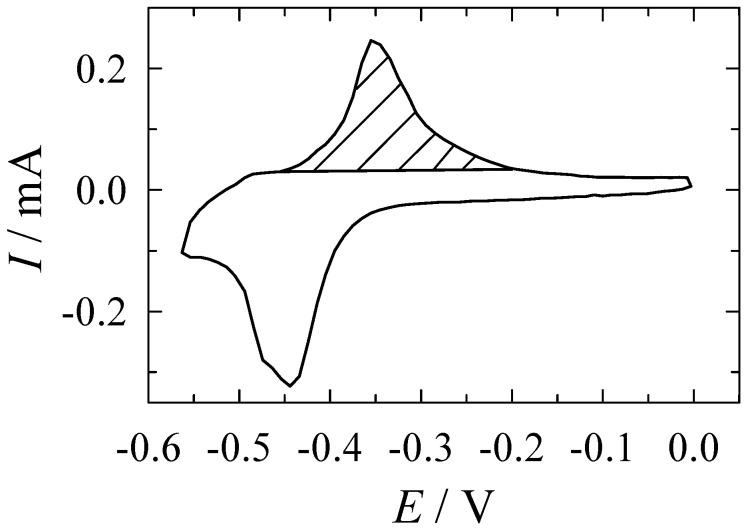
Pb underpotential deposition (UPD) on the electroless silver electrode. Solution composition (M): Pb(NO_3_)_2_—0.5 × 10^−3^, 0.1 M NaOH. Potential scan rate 50 mV s^−1^; 20 °C.

**Figure 2 materials-13-04556-f002:**
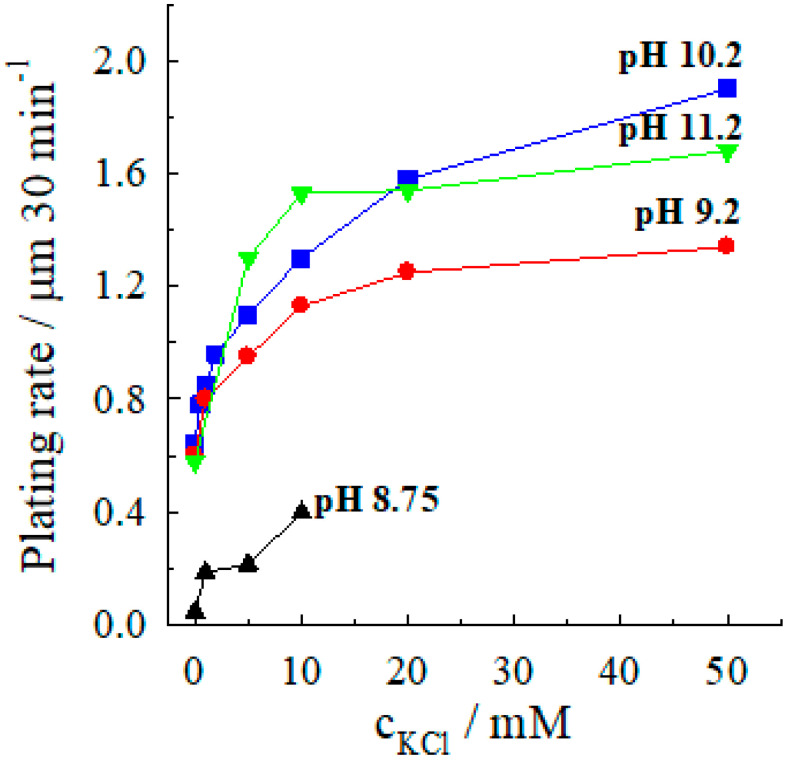
Dependence of the silver deposition rate on the chloride concentration at various pH. Solution composition (M): AgNO_3_—0.04, CoSO_4_—0.10; (NH_3_ + ${\mathrm{NH}}_{4}^{+}$)—4; 20 °C.

**Figure 3 materials-13-04556-f003:**
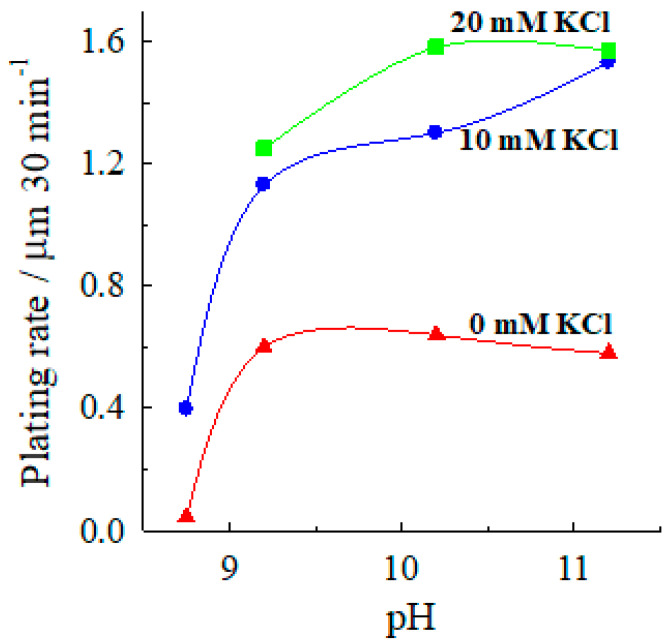
Dependence of the silver deposition rate on the solution pH at various chloride concentrations. Solution composition (M): AgNO_3_—0.04, CoSO_4_—0.10; (NH_3_ + ${\mathrm{NH}}_{4}^{+}$)—4; 20 °C.

**Figure 4 materials-13-04556-f004:**
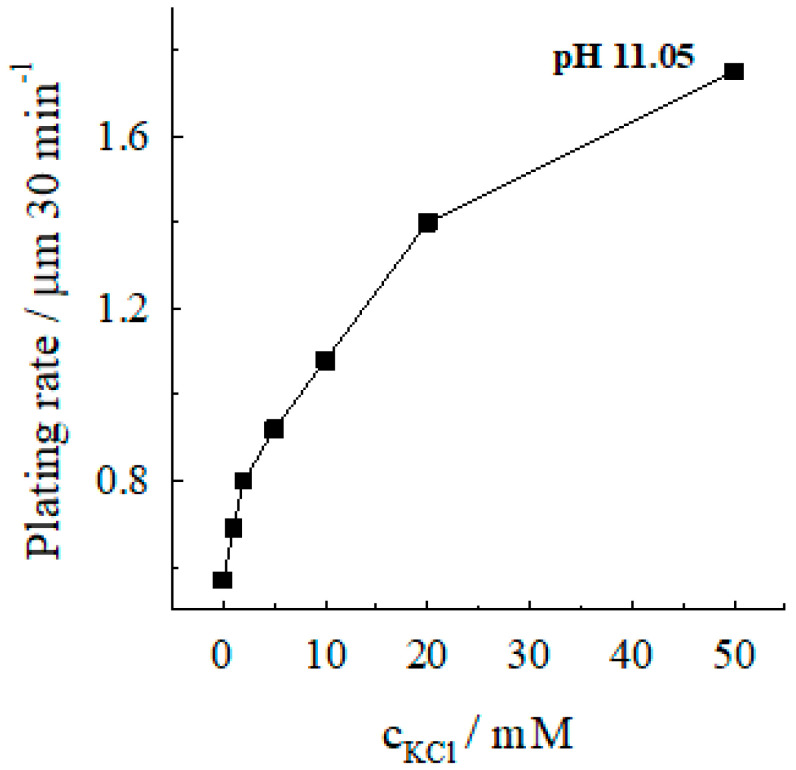
Dependence of the silver deposition rate on the chloride concentration. Solution composition (M): AgNO_3_—0.04, CoSO_4_—0.10; (NH_3_ + ${\mathrm{NH}}_{4}^{+}$)—7; pH 11.05; 20 °C.

**Figure 5 materials-13-04556-f005:**
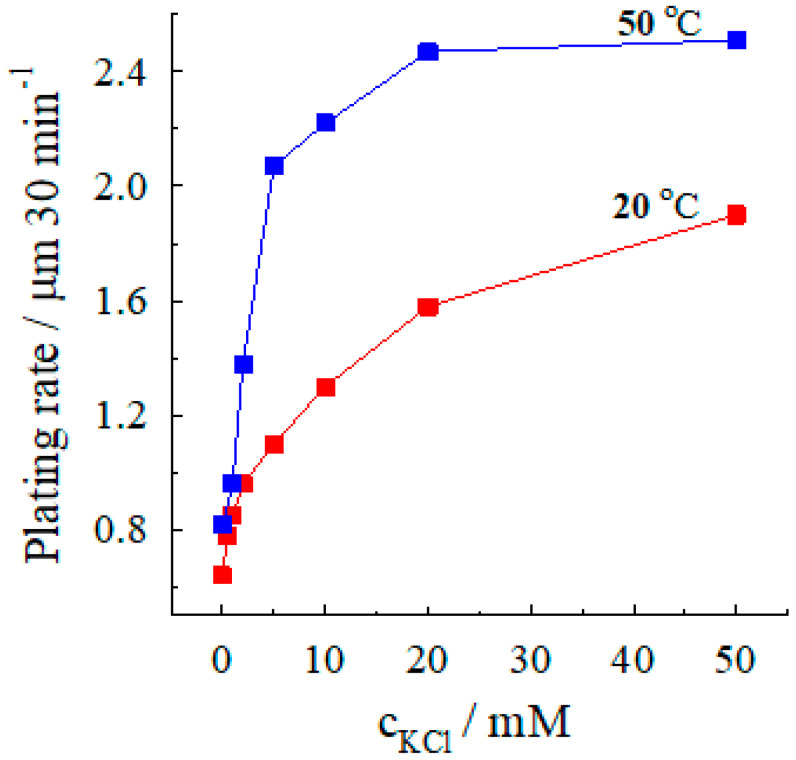
Dependence of the silver deposition rate on chloride concentration at various temperatures. Solution composition (M): AgNO_3_—0.04, CoSO_4_—0.10; (NH_3_ + ${\mathrm{NH}}_{4}^{+}$)—4; pH 10.2.

**Figure 6 materials-13-04556-f006:**
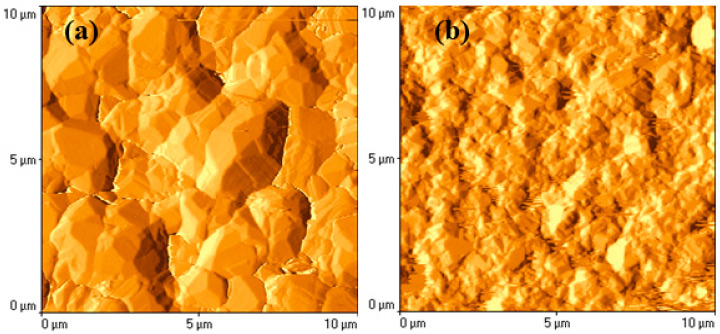
AFM images of silver coatings prepared without KCl additive (**a**) and with 100 mM KCl additive (**b**).

**Figure 7 materials-13-04556-f007:**
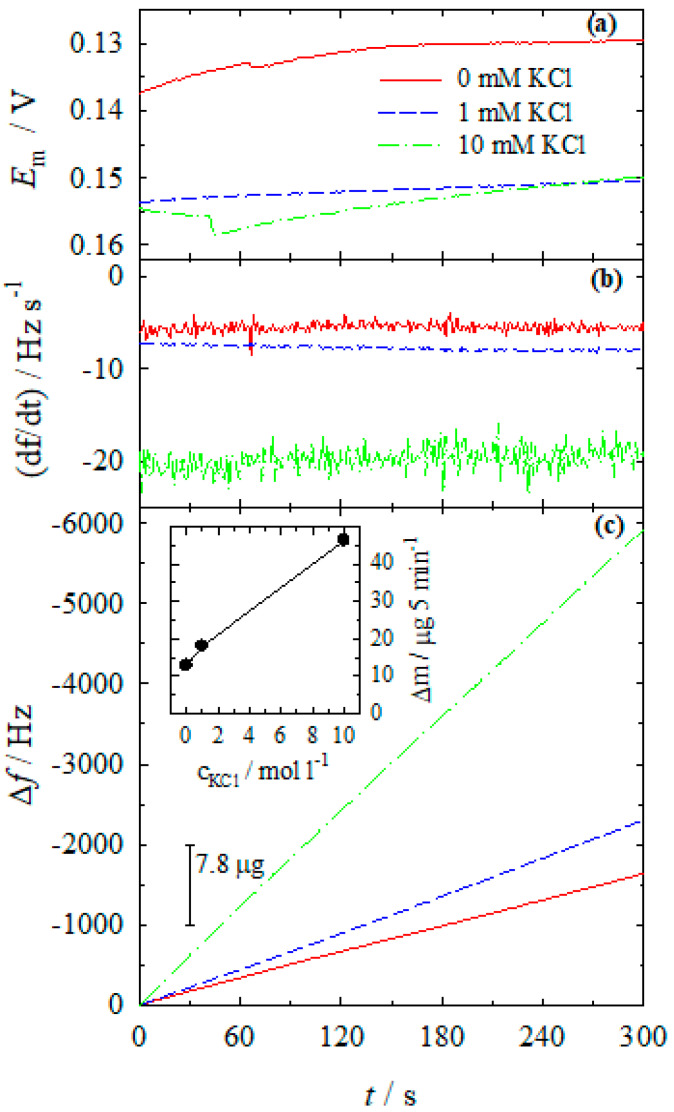
Data on electroless silver deposition on Ag electrode (0.636 cm^2^): (**a**) open-circuit potential, (**b**) frequency change rate, and (**c**) change in quartz oscillator frequency. Solution composition (M): CoSO_4_—0.1; Ag_2_SO_4_—5 × 10^−3^; KCl: 0 (solid line), 10^−3^ (dashed line), 10^−2^ (dash-dotted line); NH_3_—4.0; pH 10.9; 20 °C. The inset represents the mass gain of silver.

**Figure 8 materials-13-04556-f008:**
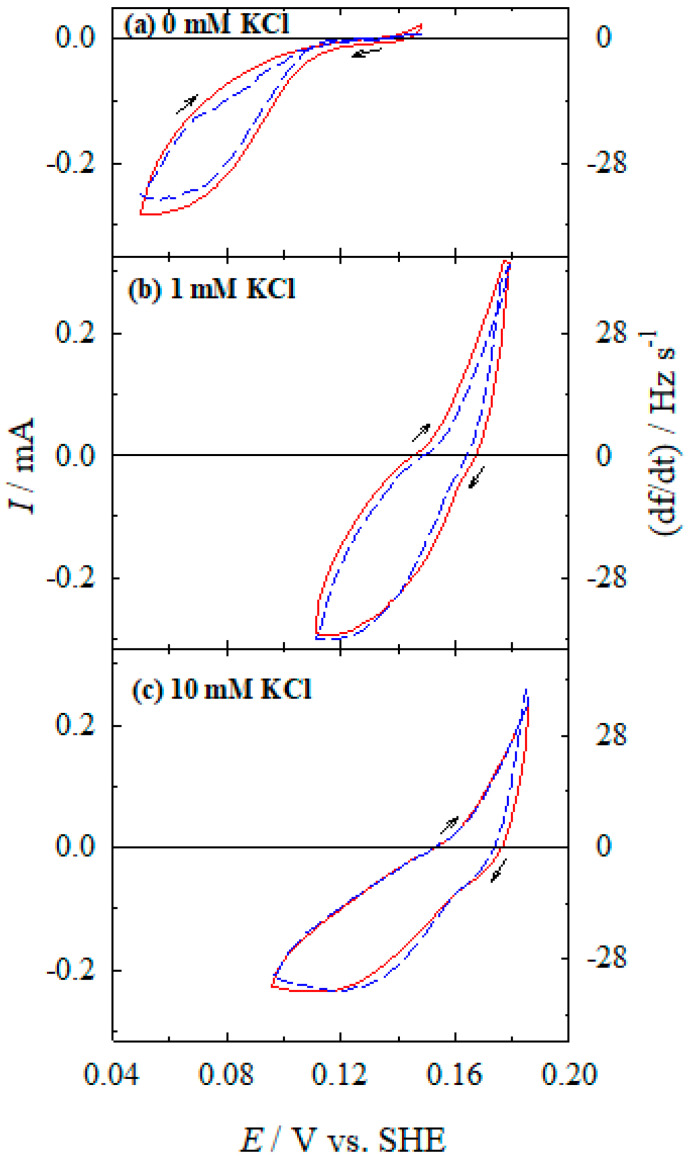
Dependences of the current (solid line) and the frequency change rate (dashed line) on the silver electrode potential. Solution composition (M): Ag_2_SO_4_—5 × 10^−3^; NH_3_—4.0; KCl: 0 (**a**), 10^−3^ (**b**), and 10^−2^ (**c**); pH 10.9; 20 °C. Potential sweep rate 2 mV s^−1^.

**Figure 9 materials-13-04556-f009:**
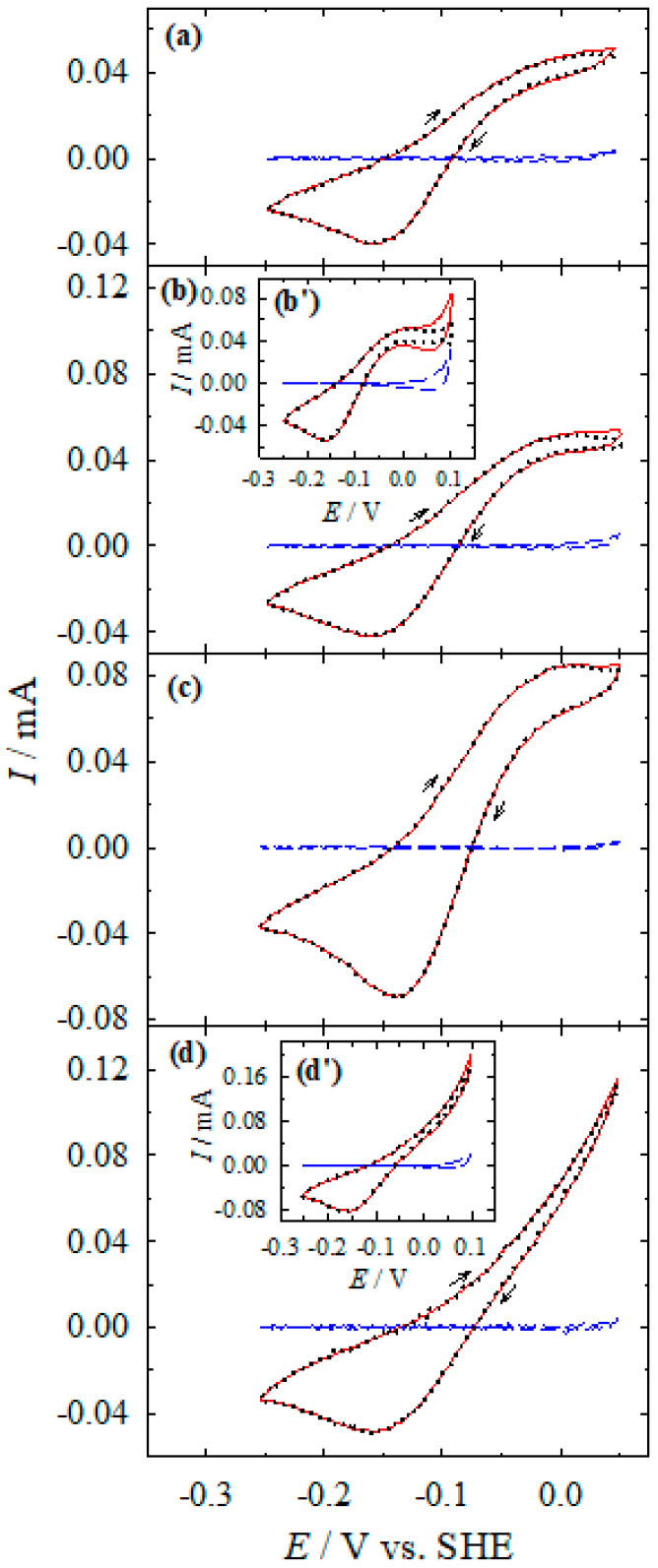
Dependencies of the current (measured directly (solid line), calculated from EQCM data (dashed line), and their difference (dotted line)) on silver electrode potential. Solution composition (M): CoSO_4_—0.1; NH_3_—4.0; KCl: 0 (**a**), 10^−3^ (**b**), 10^−2^ (**c**), and 10^−1^ (**d**); pH 10.9; 20 °C. Potential scan rate 2 mV s^−1^. The insets (**b**’) and (**d’**) represent the cyclic voltammograms (CVs) at the anodic potential limit of 0.1 V.

**Figure 10 materials-13-04556-f010:**
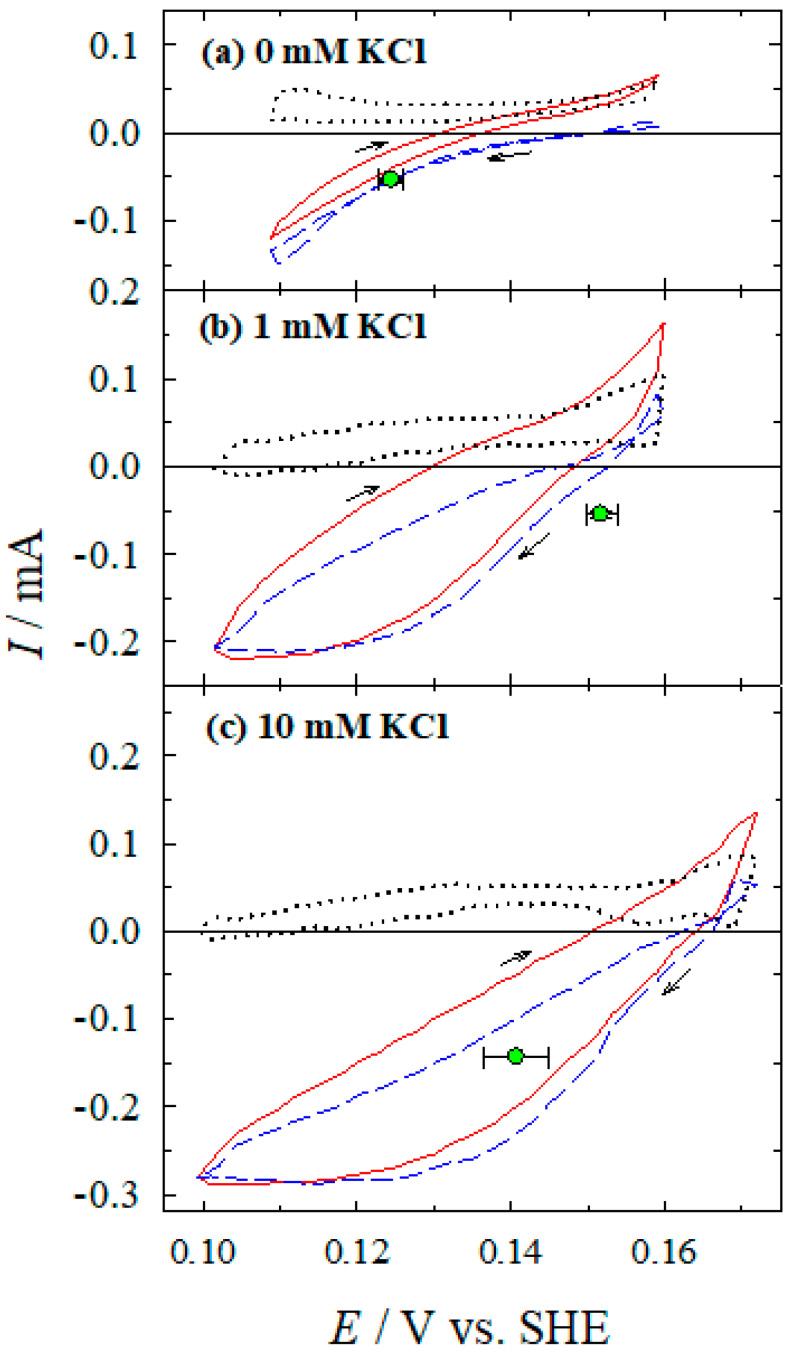
Dependences of the current (measured directly (solid line), calculated from EQCM data (dashed line), and their difference (dotted line)) on silver electrode potential in stationary solution. Solution composition (M): Ag_2_SO_4_—5 × 10^−3^; CoSO_4_—0.1; NH_3_—4.0; KCl: 0 (**a**), 10^−3^ (**b**), and 10^−2^ (**c**); pH 10.9; 20 °C. Potential sweep rate 2 mV s^−1^. ●—the rate of silver deposition under open-circuit conditions.

**Figure 11 materials-13-04556-f011:**
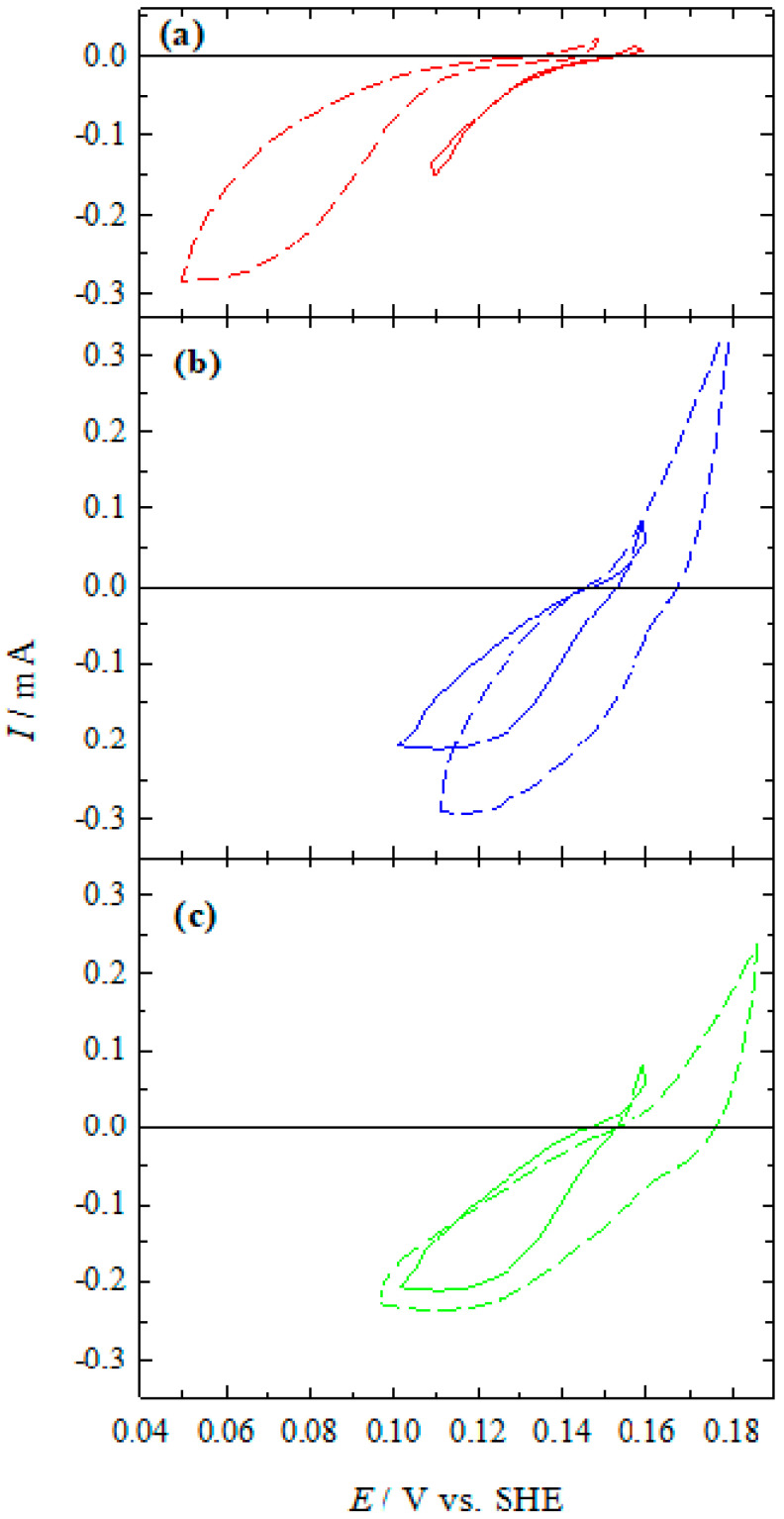
Currents for Ag deposition/dissolution are the same as in Figure 8 (dashed lines) and Figure 10 (solid lines). KCl: 0 (**a**), 10^−3^ (**b**), and 10^−2^ (**c**); pH 10.9; 20 °C. Potential sweep rate 2 mV s^−1^.

**Table 1 materials-13-04556-t001:** Composition of electroless silver-plating solutions.

	Concentration, M			
pH	AgNO_3_	CoSO_4_	NH_3_	(NH_4_)_2_SO_4_
8.75	0.04	0.10	0.50	1.75
9.2	0.04	0.10	1.00	1.50
10.2	0.04	0.10	2.50	0.75
11.2	0.04	0.10	3.80	0.10
11.5	0.04	0.10	6.00	0.50

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
