# Peer review of "Enhancing Effect of Chloride Ions on the Autocatalytic Process of Ag(I) Reduction by Co(II) Complexes"

_materials, 2020, doi:10.3390/ma13204556_

Round 1

Reviewer 1 Report

Overall, the work reads quite fluently. Results described are interesting to the research community and the discussion / interpretation of the results is sufficient.

However, I would have a few minor comments/questions/suggestions;

[1]

Line 38 

mµ > µm

[2]

Line 62

consists > consist

[3]

Line 82

for the preparing > for preparing

[4]

Line 133 + 139

'quantity of electricity'

Perhaps call this 'charge' , that would be more correct 

[5]

Line 151

details > detail

[6]

Line 158

in the most > in most

[7]

Figure 2 + 3

What is the use of these coloured 'regions' ? Is that simply wrong, due to software issues ? In any case, this obscures the figures and is not needed, I think.

[8]

Line 167

the three–fold > a three–fold

[9]

Figure 5

The line for the 50 degrees data points shows an 'indent' at 10 mM. I guess the intention was to show a smooth trend line through the data points. 

[10]

Line 188

'more developed surface'

That sounds a bit vague. Perhaps that should be re-written to clarify what is meant.

[11]

Line 218

'six-fold' 

Is that factor really correct ? It seems somewhat high.

[12]

Line 228

without of chloride > without chloride

[13]

Figure 8

I think it would be helpful to add a few arrows to indicate the scan directions of both curves. Also, the text might be extended somewhat; for instance, at what potential is the scan started, is this the first or second or .... scan. A

[14]

Line 249 - 251

Similar results were obtained by authors [20, 21]. The enhancing effect of halide ions was observed during the 250 oxidation of Co(II)–En complex on copper electrode [20, 21].

The first line 'Similar results were obtained by authors [20, 21].' is not adding much information, the second line is contain the real information. Therefore, the first line could be removed. 

[15]

Line 245-246

'At potentials more positive than 0 V simultaneously with the oxidation of Co(II) the Ag dissolution occurs.'

On the basis of which result is this conclusion drawn ? 

[16]

Line 276

'At chloride concentrations mM,'

There seems to be a concentration missing in the text.

[17]

Line 277

(<1.2 V)

I guess the authors mean: (<0.12 V)

[18]

Line 283-284

'The enhancing effect of chloride ions is related to the anodic oxidation reaction of Co(II) and
cathodic Ag(I) reduction.'

That sentence falls out of the sky, totally unclear on the basis of WHAT the authors reach to this conclusion. The sentence is stated as a fact '..... is .....'

[19]

Line 294

of a electroless > of an electroless

[20]

Line 295

'electrolytic'

Is that correct here ? Shouldn't that be 'electroless' 

[21]

Line 297

the three-fold > a three-fold

[22]

Line 299

theenhancement > the enhancement

Author Response

The authors thank the Reviewer for valuable comments. The manuscript was thoroughly revised according to the Reviewers suggestions. In general, the title of the manuscript was change, the Figures 7 and 9 were revised, and a new Figure 11 was additionally added to the revised version of the manuscript.

Reviewer: However, I would have a few minor comments/questions/suggestions;

[1] Line 38 

mµ > µm

[2] Line 62

consists > consist

[3] Line 82

for the preparing > for preparing

[4] Line 133 + 139

'quantity of electricity'

Perhaps call this 'charge' , that would be more correct 

[5] Line 151

details > detail

[6] Line 158

in the most > in most

[8] Line 167

the three–fold > a three–fold

[12]

Line 228

without of chloride > without chloride

[19] Line 294

of a electroless > of an electroless

[20] Line 295

'electrolytic'

Is that correct here ? Shouldn't that be 'electroless' 

[21] Line 297

the three-fold > a three-fold

[22] Line 299

theenhancement > the enhancement

[17] Line 277

(<1.2 V)

I guess the authors mean: (<0.12 V)

Authors: Thank the reviewer for valuable comments/suggestions. The all mistakes were revised according to the Reviewer suggestions.

Reviewer: [7] Figure 2 + 3

What is the use of these coloured 'regions' ? Is that simply wrong, due to software issues ? In any case, this obscures the figures and is not needed, I think.

Authors: The colored regions were obtained during the converting doc file to pdf due to software issues. We believe that in the final manuscript version this problem will be solved. Real Figures 2 and 3 see below.

Figs. 2 and 3.

Reviewer: [9] Figure 5

The line for the 50 degrees data points shows an 'indent' at 10 mM. I guess the intention was to show a smooth trend line through the data points.

Authors: We decline the spline line of the curve at 50 oC. The figure 5 was corrected.

Reviewer: [10] Line 188

'more developed surface'

That sounds a bit vague. Perhaps that should be re-written to clarify what is meant.

Authors: The text was re-written.

Reviewer: [11] Line 218

'six-fold' 

Is that factor really correct ? It seems somewhat high.

Authors: The Reviewer is correct. This factor was revised.

Reviewer:  [13] Figure 8

I think it would be helpful to add a few arrows to indicate the scan directions of both curves. Also, the text might be extended somewhat; for instance, at what potential is the scan started, is this the first or second or .... scan. A

Authors: The figure was revised by adding the scan directions and the data were discussed in the text.

Reviewer: [14] Line 249 - 251

Similar results were obtained by authors [20, 21]. The enhancing effect of halide ions was observed during the 250 oxidation of Co(II)–En complex on copper electrode [20, 21].

The first line 'Similar results were obtained by authors [20, 21].' is not adding much information, the second line is contain the real information. Therefore, the first line could be removed. 

Authors: It was revised – the sentence was removed.

Reviewer: [15] Line 245-246

'At potentials more positive than 0 V simultaneously with the oxidation of Co(II) the Ag dissolution occurs.'

On the basis of which result is this conclusion drawn ? 

Authors: The Figure 9 was revised by adding the insets (b’ and d’) with CVs with higher anodic potential limit.

Reviewer: [16] Line 276

'At chloride concentrations mM,'

There seems to be a concentration missing in the text.

Authors: It was revised.

Reviewer: [18] Line 283-284

'The enhancing effect of chloride ions is related to the anodic oxidation reaction of Co(II) and
cathodic Ag(I) reduction.'

That sentence falls out of the sky, totally unclear on the basis of WHAT the authors reach to this conclusion. The sentence is stated as a fact '..... is .....'

Authors: The enhancing effect of chloride ions is related mainly with the enhancing of Co(II) oxidation to Co(III). The text was re-written.

Reviewer 2 Report

In a study of a electroless silver deposition system with a reducing agent Co(II)ammonia complexes, it was found that the addition of chlorides accelerates the overall process of electrolytic silver deposition.- the presented conclusion is interesting, but on the way to it, a few things need to be corrected and explained:

1)Very long abstract - and not very specific. The conclusions, on the other hand, are too short, because they do not emphasize the novelty !!!

2)In the introduction, little information is provided on what we can read about the problem under consideration and what is completely new

3)What is fig 1 supposed to contribute to the experimental part?

4) Figures 2 and 3 are not clear - please describe exactly what they bring?

5)fig 6 sloppy pasted picture frame why should it be used?

6)Fig 8 is wrongly described, the interpretation of the drawing is wrong, please compare it with the literature

After corrections the work can be published

Author Response

The authors thank the Reviewer for valuable comments. The manuscript was thoroughly revised according to the Reviewers suggestions. In general, the title of the manuscript was change, the Figures 7 and 9 were revised, and a new Figure 11 was additionally added to the revised version of the manuscript.

In a study of a electroless silver deposition system with a reducing agent Co(II)–ammonia complexes, it was found that the addition of chlorides accelerates the overall process of electrolytic silver deposition.- the presented conclusion is interesting, but on the way to it, a few things need to be corrected and explained:

Reviewer: 1) Very long abstract - and not very specific. The conclusions, on the other hand, are too short, because they do not emphasize the novelty !!!

Authors: The abstract was shortened. Conclusions were revised.

Reviewer: 2) In the introduction, little information is provided on what we can read about the problem under consideration and what is completely new

Authors: It was revised.

Reviewer: 3) What is fig 1 supposed to contribute to the experimental part?

Authors: This figure clarify the area for calculation of charge.

Reviewer: 4) Figures 2 and 3 are not clear - please describe exactly what they bring?

Authors: The colored regions were obtained during the converting doc file to pdf due to software issues. We believe that in the final manuscript version this problem will be solved. Real Figures 2 and 3 see below.

Figs. 2 and 3.

Reviewer: 5) fig 6 sloppy pasted picture frame why should it be used?

Authors: It was corrected.

Reviewer: 6) Fig 8 is wrongly described, the interpretation of the drawing is wrong, please compare it with the literature

Authors: The text was revised and re-written.

Reviewer 3 Report

The manuscript "Enhancing effect of chloride ions on the autocatalytic process of Ag(I) reduction by Co(II) complexes" has been reviewed.

The effect of halide additive, namely chloride ions, on the rate of electroless silver deposition was investigated by the authors by electrochemical and chemical kinetics methods.

The manuscript is interesting and well organized.

The topic is focused on the deposition rate. By the way a short dissertation on the adhesion properties and mechanical properties (i.e. Scratch test, not only surface roughness) of the manufactured coating would be of interest for the readers.

At the end in the following some minor corrections are reported:

Line 38 "micron" not (mμ);

Line 44: please supply more details about the citations [8-14], specifying the one by one the reason of each citation;

Line 62: too many spaces after "bulk".

Author Response

The authors thank the Reviewer for valuable comments. The manuscript was thoroughly revised according to the Reviewers suggestions. In general, the title of the manuscript was change, the Figures 7 and 9 were revised, and a new Figure 11 was additionally added to the revised version of the manuscript.

Reviewer: The topic is focused on the deposition rate. By the way a short dissertation on the adhesion properties and mechanical properties (i.e. Scratch test, not only surface roughness) of the manufactured coating would be of interest for the readers.

Authors: Adhesion of silver coatings is sufficient – the coatings obtained pass the scotch tape test.

Reviewer: At the end in the following some minor corrections are reported:

Line 38 "micron" not (mμ);

Authors: It was corrected.

Reviewer: Line 44: please supply more details about the citations [8-14], specifying the one by one the reason of each citation;

Authors: The text was re-written and specified.

Reviewer: Line 62: too many spaces after "bulk".

Authors: It was corrected.

Reviewer 4 Report

This contribution describes the process of the electroless silver plating with halide additive. Particularly, the addition of chlorides can promote the process of electrolytic silver deposition when the three-fold acceleration. And currently it reads like a routine study and needs some major surgery to move into the significant bracket. Some specific comments are shown below:

  1. For the effect of halide additive, the author can compare different halogen atoms to explain the difference.
  2. Is it uniform for the coating of silver atoms? If not, does it affect the rate?

Author Response

The authors thank the Reviewer for valuable comments. The manuscript was thoroughly revised according to the reviewers' suggestions. In general, the title of the manuscript was changed, Figures 7 and 9 were revised, and a new Figure 11 was additionally added to the revised version of the manuscript.

Some specific comments are shown below:

Reviewer: For the effect of halide additive, the author can compare different halogen atoms to explain the difference.

Authors: In the study, we don’t use other halides than chloride.

Reviewer: Is it uniform for the coating of silver atoms? If not, does it affect the rate?

Authors: In general, the coatings are uniform having a regular crystalline structure.

Reviewer 5 Report

The authors present data exploring the effect of chloride on the rate of silver electroless deposition from a solution containing a Co(II) complex as the reducing agent. The rate of deposition was measured directly by ex-situ weighing, as well as by using electrochemical quartz crystal microbalance (EQCM) data. By comparing electrochemical current to EQCM frequency shifts, the rate for each partial reaction involved in the electroless deposition was determined. Addition of chloride increases the silver deposition rate for the system. The results are interesting and will be useful to the community. This work is suitable for publication in Materials after the comments below are considered.

MAJOR COMMENTS

  1. The conclusions for this work seem reasonable; however the presentation is a bit unclear. An additional figure or two directly comparing, the Ag partial reactions with and without Co (from Figs 8 and 10) and the Co partial reactions with and without Ag (from Figs. 9 and 10) would help to emphasize the points that are made at the end of the manuscript.
  2. The lateral size of the features in the AFM images (Fig 6) are very different between the samples with and without Cl, but no comments are made on these changes.

MINOR COMMENTS

  1. There are two Sections labeled 2.3, and the second one (about the coatings characterization) would be more clear if placed before the first one (about the electrochemical and EQCM measurements). The characterization described is for the samples fabricated with the method described in Section 2.2. Placing additional information between the two parts breaks the logical flow of the material. Section 2.4 may also be more clear if put before the second Section 2.3.
  2. Some figures should be made more clear. For example, there is no reason for the solid fill colors in Figs 2 and 3. Additionally, a height color bar should be included for the AFM images in Fig 6.
  3. The Results and Discussion section (Section 3) could be broken up into sub-sections for greater clarity.
  4. The caption for Fig 7 does not describe the data in the inset. (a’) and (b’). Additionally, is there a reason why the vertical Em axis in (a) and (b’) are flipped with respect to each other?
  5. For the paragraphs describing the CV and EQM data using various solutions (lines 220 and following), each solution type (Ag only, Co only, Ag and Co) should be clarified as the results are being described.
  6. The entire manuscript has a number of small wording and grammatical errors that can be corrected by a careful editing process.

Author Response

The authors thank the Reviewer for valuable comments. The manuscript was thoroughly revised according to the reviewers’ suggestions. In general, the title of the manuscript was changed, Figures 7 and 9 were revised, and a new Figure 11 was additionally added to the revised version of the manuscript.

MAJOR COMMENTS

Reviewer: The conclusions for this work seem reasonable; however the presentation is a bit unclear. An additional figure or two directly comparing, the Ag partial reactions with and without Co (from Figs 8 and 10) and the Co partial reactions with and without Ag (from Figs. 9 and 10) would help to emphasize the points that are made at the end of the manuscript.

Authors: The additional Figure 11 was added for comparison purposes and discussed in the text. 

Reviewer: The lateral size of the features in the AFM images (Fig 6) are very different between the samples with and without Cl, but no comments are made on these changes.

Authors: The additional comments were added. 

MINOR COMMENTS

Reviewer: There are two Sections labeled 2.3, and the second one (about the coatings characterization) would be more clear if placed before the first one (about the electrochemical and EQCM measurements). The characterization described is for the samples fabricated with the method described in Section 2.2. Placing additional information between the two parts breaks the logical flow of the material. Section 2.4 may also be more clear if put before the second Section 2.3.

Authors: The Experimental part was revised according to the reviewer's suggestions.

Reviewer: Some figures should be made more clear. For example, there is no reason for the solid fill colors in Figs 2 and 3. Additionally, a height color bar should be included for the AFM images in Fig 6.

Authors: The colored regions were obtained during the converting doc file to pdf due to software issues. We believe that in the final manuscript version this problem will be solved. Real Figures 2 and 3 see below.

Figs. 2 and 3.

Reviewer: The Results and Discussion section (Section 3) could be broken up into sub-sections for greater clarity.

Authors: The Results and Discussion section (Section 3) was broken up into three sub-sections.

Reviewer: The caption for Fig 7 does not describe the data in the inset. (a’) and (b’). Additionally, is there a reason why the vertical Em axis in (a) and (b’) are flipped with respect to each other?

Authors: Figure 7 and the caption were revised.

Reviewer: For the paragraphs describing the CV and EQM data using various solutions (lines 220 and following), each solution type (Ag only, Co only, Ag and Co) should be clarified as the results are being described.

Authors: It was revised.

Reviewer: The entire manuscript has a number of small wording and grammatical errors that can be corrected by a careful editing process.

Authors: It was corrected.

Round 2

Reviewer 3 Report

All the issues have been addressed.

Accept in the present form.

Reviewer 4 Report

I think this manuscript should be accepted as it is.